# Fixing the train-test resolution discrepancy

**Hugo Touvron, Andrea Vedaldi, Matthijs Douze, Hervé Jégou**

Facebook AI Research

## Abstract

Data-augmentation is key to the training of neural networks for image classification. This paper first shows that existing augmentations induce a significant discrepancy between the size of the objects seen by the classifier at train and test time: in fact, a lower train resolution improves the classification at test time!

We then propose a simple strategy to optimize the classifier performance, that employs different train and test resolutions. It relies on a computationally cheap fine-tuning of the network at the test resolution. This enables training strong classifiers using small training images, and therefore significantly reduce the training time. For instance, we obtain 77.1% top-1 accuracy on ImageNet with a ResNet-50 trained on 128×128 images, and 79.8% with one trained at 224×224.

A ResNeXt-101 32x48d pre-trained with weak supervision on 940 million 224×224 images and further optimized with our technique for test resolution 320×320 achieves 86.4% top-1 accuracy (top-5: 98.0%). To the best of our knowledge this is the highest ImageNet single-crop accuracy to date.

## 1 Introduction

Convolutional Neural Networks [18] (CNNs) are used extensively in computer vision tasks such as image classification [17], object detection [27], inpainting [37], style transfer [9] and even image compression [28]. In order to obtain the best possible performance from these models, the training and testing data distributions should match. However, often data pre-processing procedures are different for training and testing. For instance, in image recognition the current best training practice is to extract a rectangle with random coordinates from the image, to artificially increase the amount of training data. This region, which we call the *Region of Classification* (RoC), is then resized to obtain an image, or crop, of a fixed size (in pixels) that is fed to the CNN. At test time, the RoC is instead set to a square covering the central part of the image, which results in the extraction of a *center crop*. This reflects the bias of photographers who tend center important visual content. Thus, while the crops extracted at training and test time have the same size, they arise from different RoCs, which skews the distribution of data seen by the CNN.

Over the years, training and testing pre-processing procedures have evolved to improve the performance of CNNs, but so far they have been optimized separately [7]. In this paper, we first show that this separate optimization has led to a significant distribution shift between training and testing regimes with a detrimental effect on the test-time performance of models. We then show that this problem can be solved by jointly optimizing the choice of resolutions and scales at training and test time, while keeping the same RoC sampling. Our strategy only requires to fine-tune two layers in order to compensate for the shift in statistics caused by changing the crop size. This retains the advantages of existing pre-processing protocols for training and testing, including augmenting the training data, while compensating for the distribution shift.

Our approach is based on a rigorous analysis of the effect of pre-processing on the statistics of natural images, which shows that increasing the size of the crops used at test time compensates for randomly

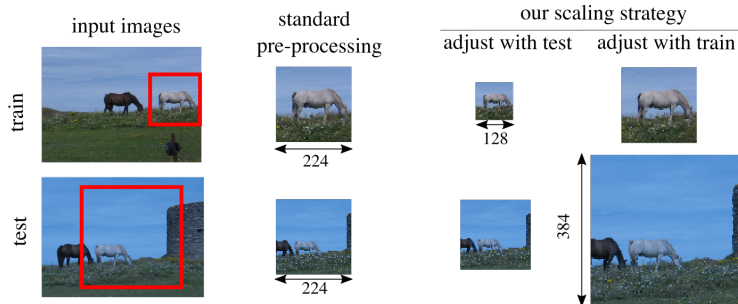

Figure 1: Selection of the image regions fed to the network at training time and testing time, with typical data-augmentation. The red region of classification is resampled as a crop that is fed to the neural net. For objects that have as similar size in the input image, like the white horse, the standard augmentations typically make them larger at training time than at test time (second column). To counter this effect, we either reduce the train-time resolution, or increase the test-time resolution (third and fourth column). The horse then has the same size at train and test time, requiring less scale invariance for the neural net. Our approach only needs a computationally cheap fine-tuning.

sampling the RoCs at training time. This analysis also shows that we need to use lower resolution crops at training than at test time. This significantly impacts the processing time: halving the crop resolution leads to a threefold reduction in the network evaluation speed and reduces significantly the memory consumption for a typical CNN, which is especially important for training on GPUs. For instance, for a target test resolution of $224 \times 224$, training at resolution $160 \times 160$ provides better results than the standard practice of training at resolution $224 \times 224$, while being more efficient. In addition we can adapt a ResNet-50 train at resolution $224 \times 224$ for the test resolution $320 \times 320$ and thus obtain top-1 accuracy of 79.8% (single-crop) on ImageNet.

Alternatively, we leverage the improved efficiency to train high-accuracy models that operate at much higher resolution at test time while still training quickly. For instance, we achieve an top-1 accuracy of 86.4% (single-crop) on ImageNet with a ResNeXt-101 32x48d pre-trained in weakly-supervised fashion on 940 million public images. Finally, our method makes it possible to save GPU memory, which we exploit in the optimization: employing larger batch sizes leads to a better final performance [13].

## 2   Related work

**Image classification**   is a core problem in computer vision, and used as a benchmark task by the community to measure progress on image understanding. Models pre-trained for image classification, usually on the ImageNet database [8], transfer to a variety of other applications [24]. Furthermore, advances in image classification translate to improved results on many other tasks [10, 15].

Recent research in image classification has demonstrated improved performance by considering larger networks and higher resolution images [14, 22]. For instance, the state of the art in the ImageNet ILSVRC 2012 benchmark is currently held by the ResNeXt-101 32x48d [22] architecture with 829M parameters using $224 \times 224$ images for training. The state of the art for a model learned from scratch is currently held by the EfficientNet-b7 [34] with 66M parameters using $600 \times 600$ images for training. In this paper, we focus on the ResNet-50 architecture [11] due to its good accuracy/cost tradeoff (25.6M parameters) and its popularity. We also conduct some experiments using the PNASNet-5-Large [21] architecture (86.1M parameters), which exhibits good performance on ImageNet with a reasonable training time, and with the ResNeXt-101 32x48d [22] weakly supervised, which is best publicly available network on ImageNet.

**Data augmentation**   is routinely employed at training time to improve model generalization and reduce overfitting. Typical transformations [3, 5, 32] include: random-size crop, horizontal flip and color jitter. In our paper, we adopt the standard set of augmentations commonly used in image classification. As a reference, we consider the default models in the PyTorch library. The accuracy is also improved by combining multiple data augmentations at test time, although this means that sev-

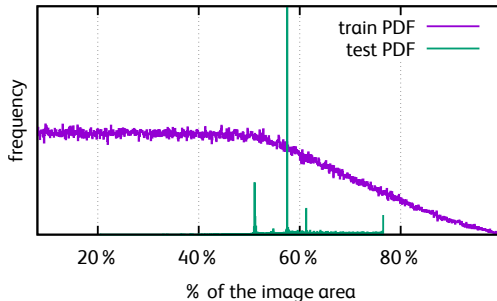

Figure 2: Empirical distribution of the areas of the RoCs as a fraction of the image areas extracted by data augmentation. The data augmentation schemes are the standard ones used at training and testing time for CNN classifiers. The spiky distribution at test time is due to the fact that RoCs are center crops and the only remaining variability is due to the different image aspect ratios. Notice that the distribution is very different at training and testing time.

eral forward passes are required to classify one image. For example, [11, 17, 32] used ten crops (one central, and one for each corner of the image and their mirrored versions). Another performance-boosting strategy is to classify an image by feeding it at multiple resolutions [11, 30, 32], again averaging the predictions. More recently, multi-scale strategies such as the feature pyramid network [20] have been proposed to directly integrate multiple resolutions in the network, both at train and test time, with significant gains in category-level detection.

**Feature pooling.** A recent approach [5] employs $p$-pooling instead of average pooling to adapt the network to test resolutions significantly higher than the training resolution. The authors show that this improves the network's performance, in accordance with the conclusions drawn by Boureau *et al.* [6]. Similar pooling techniques have been employed in image retrieval for a few years [26, 35], where high-resolution images are required to achieve a competitive performance.

## 3    Region selection and scale statistics

Applying a Convolutional Neural Network (CNN) classifier to an image generally requires to pre-process the image. One of the key steps involves selecting a rectangular region in the input image, which we call *Region of Classification* (RoC). The RoC is then extracted and resized to a square crop of a size compatible with the CNN, e.g., AlexNet requires a $224 \times 224$ crop as input.

While this process is simple, in practice it has two subtle but significant effects on how the image data is presented to the CNN. First, the resizing operation changes the *apparent size* of the objects in the image (section 3.1). This is important because CNNs *do not* have a predictable response to a scale change (as opposed to translations). Second, the choice of different crop sizes (for architectures such as ResNet that admit non-fixed inputs) has an effect on the *statistics* of the network activations, especially after global pooling layers (section 3.2). This section analyses in detail these two effects. In the discussion, we use the following conventions: The "input image" is the original training or testing image; the RoC is a rectangle in the input image; and the "crop" is the pixels of the RoC, rescaled with bilinear interpolation to a fixed resolution, then fed to the CNN.

### 3.1    Scale and apparent object size

If a CNN is to acquire a scale-invariant behavior for object recognition, it must *learn* it from data. However, resizing the input images in pre-processing changes the distribution of objects sizes. Since different pre-processing protocols are used at training and testing time[1], the size distribution *differs* in the two cases. This is quantified next.

### 3.1.1    Relation between apparent and actual object sizes

We consider the following imaging model: the camera projects the 3D world onto a 2D image, so the apparent size of the objects is inversely proportional to their distance from the camera. For simplicity, we model a 3D object as an upright square of height and width $R \times R$ at a distance $Z$ from the camera, and fronto-parallel to it. Hence, its image is a $r \times r$ rectangle, where the *apparent*

*size* $r$ is given by $r = fR/Z$ where $f$ is the *focal length* of the camera. Thus we can express the apparent size as the product $r = f \cdot r_1$ of the focal length $f$, which depends on the camera, and of the variable $r_1 = R/Z$, whose distribution $p(r_1)$ is camera-independent. While the focal length is variable, the *field of view* angle $\theta_{\text{FOV}}$ of most cameras is usually in the $[40°, 60°]$ range. Hence, for an image of size $H \times W$ one can write $f = k\sqrt{HW}$ where $k^{-1} = 2\tan(\theta_{\text{FOV}}/2) \approx 1$ is approximately constant. With this definition for $f$, the apparent size $r$ is expressed in pixels.

### 3.1.2 Effect of image pre-processing on the apparent object size

Now, we consider the effect of rescaling images on the apparent size of objects. If an object has an extent of $r \times r$ pixels in the input image, and if $s$ is the scaling factor between input image and the crop, then by the time the object is analysed by the CNN, it will have the new size of $rs \times rs$ pixels. The scaling factor $s$ is determined by the pre-processing protocol, discussed next.

**Train-time scale augmentation.** As a prototypical augmentation protocol, we consider `RandomResizedCrop` in PyTorch, which is very similar to augmentations used by other toolkits such as Caffe and the original AlexNet. `RandomResizedCrop` takes as input an $H \times W$ image, selects a RoC at random, and resizes the latter to output a $K_{\text{train}} \times K_{\text{train}}$ crop. The RoC extent is obtained by first sampling a scale parameter $\sigma$ such that $\sigma^2 \sim U([\sigma_-^2, \sigma_+^2])$ and an aspect ratio $\alpha$ such that $\ln\alpha \sim U([\ln\alpha_-, \ln\alpha_+])$. Then, the size of the RoC in the input image is set to $H_{\text{RoC}} \times W_{\text{RoC}} = \sqrt{\sigma\alpha HW} \times \sqrt{\sigma HW/\alpha}$. The RoC is resized anisotropically with factors $(K_{\text{train}}/H_{\text{RoC}}, K_{\text{train}}/W_{\text{RoC}})$ to generate the output image. Assuming for simplicity that the input image is square (i.e. $H = W$) and that $\alpha = 1$, the scaling factor from input image to output crop is given by:

$$s = \frac{\sqrt{K_{\text{train}}K_{\text{train}}}}{\sqrt{H_{\text{RoC}}W_{\text{RoC}}}} = \frac{1}{\sigma} \cdot \frac{K_{\text{train}}}{\sqrt{HW}}. \tag{1}$$

By scaling the image in this manner, the apparent size of the object becomes

$$r_{\text{train}} = s \cdot r = sf \cdot r_1 = \frac{kK_{\text{train}}}{\sigma} \cdot r_1. \tag{2}$$

Since $kK_{\text{train}}$ is constant, differently from $r$, $r_{\text{train}}$ does *not* depend on the size $H \times W$ of the input image. Hence, pre-processing *standardizes* the apparent size, which otherwise would depend on the input image resolution. This is important as networks do not have built-in scale invariance.

**Test-time scale augmentation.** As noted above, test-time augmentation usually differs from train-time augmentation. The former usually amounts to: isotropically resizing the image so that the shorter dimension is $K_{\text{test}}^{\text{image}}$ and then extracting a $K_{\text{test}} \times K_{\text{test}}$ crop (`CenterCrop`) from that. Under the assumption that the input image is square ($H = W$), the scaling factor from input image to crop rewrites as $s = K_{\text{test}}^{\text{image}}/\sqrt{HW}$, so that

$$r_{\text{test}} = s \cdot r = kK_{\text{test}}^{\text{image}} \cdot r_1. \tag{3}$$

This has a a similar size standardization effect as the train-time augmentation.

**Lack of calibration.** Comparing eqs. (2) and (3), we conclude that the same input image containing an object of size $r_1$ results in two different apparent sizes if training or testing pre-processing is used. These two sizes are related by:

$$\frac{r_{\text{test}}}{r_{\text{train}}} = \sigma \cdot \frac{K_{\text{test}}^{\text{image}}}{K_{\text{train}}}. \tag{4}$$

In practice, for standard networks such as AlexNet $K_{\text{test}}^{\text{image}}/K_{\text{train}} \approx 1.15$; however, the scaling factor $\sigma$ is sampled (with the square law seen above) in a range $[\sigma_-, \sigma_+] = [0.28, 1]$. Hence, at testing time the same object may appear as small as a third of what it appears at training time. For standard values of the pre-processing parameters, the expected value of this ratio w.r.t. $\sigma$ is

$$\mathrm{E}\left[\frac{r_{\text{test}}}{r_{\text{train}}}\right] = F \cdot \frac{K_{\text{test}}^{\text{image}}}{K_{\text{train}}} \approx 0.80, \qquad F = \frac{2}{3} \cdot \frac{\sigma_+^3 - \sigma_-^3}{\sigma_+^2 - \sigma_-^2}, \tag{5}$$

where $F$ captures all the sampling parameters.

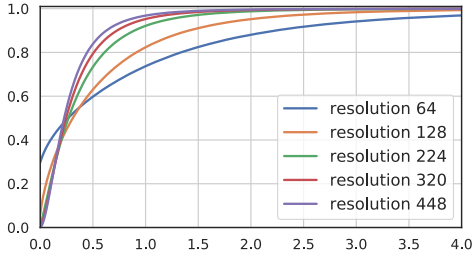

Figure 3: Cumulative density function of the vectors components on output of the spatial average pooling operator, for a standard ResNet-50 trained at resolution 224, and tested at different resolutions. The distribution is measured on the validation images of Imagenet.

## 3.2 Scale and activation statistics

In addition to affecting the apparent size of objects, pre-processing also affects the activation statistics of the CNN, especially if its architecture allows changing the size of the input crop. We first look at the *receptive field size* of a CNN activation in the previous layer. This is the number of input spatial locations that affect that response. For the convolutional part of the CNN, comprising linear convolution, subsampling, ReLU, and similar layers, changing the input crop size is almost neutral because the receptive field is unaffected by the input size. However, for classification the network must be terminated by a pooling operator (usually average pooling) in order to produce a fixed-size vector. Changing the size of the input crop strongly affects the activation statistics of this layer.

**Activation statistics.** We measure the distribution of activation values after the average pooling in a ResNet-50 in fig. 3. As it is applied on a ReLU output, all values are non-negative. At the default crop resolution of $K_{test} = K_{train} = 224$ pixels, the activation map is $7 \times 7$ with a depth of 2048. At $K_{test} = 64$, the activation map is only $2 \times 2$: pooling only 0 values becomes more likely and activations are more sparse (the rate of 0's increases form 0.5% to 29.8%). The values are also more spread out: the fraction of values above 2 increases from 1.2% to 11.9%. Increasing the resolution reverts the effect: with $K_{test} = 448$, the activation map is $14 \times 14$, the output is less sparse and less spread out.

This simple statistical observations shows that if the distribution of activations changes at test time, the values are not in the range that the final classifier layers (linear & softmax) were trained for.

## 3.3 Larger test crops result in better accuracy

Despite the fact that increasing the crop size affects the activation statistics, it is generally beneficial for accuracy, since as discussed before it reduces the train-test object size mismatch. For instance, the accuracy of ResNet-50 on the ImageNet validation set as $K_{test}$ is changed (see section 5) are:

| $K_{test}$ | 64 | 128 | 224 | 256 | 288 | 320 | 384 | 448 |
|---|---|---|---|---|---|---|---|---|
| accuracy | 29.4 | 65.4 | 77.0 | 78.0 | 78.4 | 78.3 | 77.7 | 76.6 |

Thus for $K_{test} = 288$ the accuracy is 78.4%, which is *greater* than 77.0% obtained for the native crop size $K_{test} = K_{train} = 224$ used in training. In fig. 5, we see this result is general: better accuracy is obtained with higher resolution crops at test time than at train time. In the next section, we explain and leverage this discrepancy by adjusting the network's weights.

## 4 Method

Based on the analysis of section 3, we propose two improvements to the standard setting. First, we show that the difference in apparent object sizes at training and testing time can be removed by increasing the crop size at test time, which explains the empirical observation of section 3.3. Second, we slightly adjust the network before the global average pooling layer in order to compensate for the change in activation statistics due to the increased size of the input crop.

### 4.1 Calibrating the object sizes by adjusting the crop size

Equation (5) estimates the change in the apparent object sizes during training and testing. If the size of the intermediate image $K_{test}^{image}$ is *increased* by a factor $\alpha$ (where $\alpha \approx 1/0.80 = 1.25$ in the example) then at test time, the apparent size of the objects is increased by the same factor. This

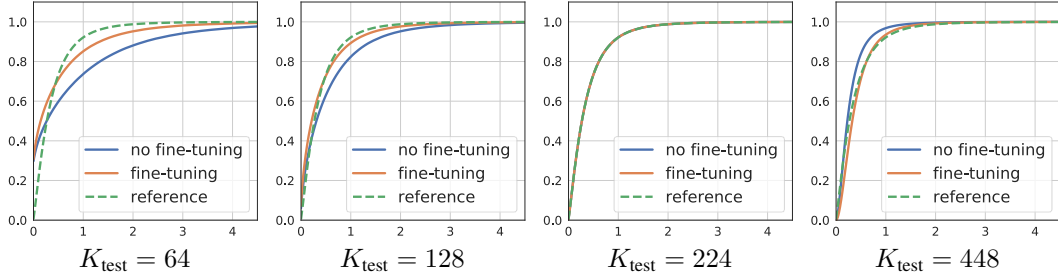

Figure 4: CDF of the activations on output of the average pooling layer, for a ResNet-50, when tested at different resolutions $K_{\text{test}}$. Compare the state before and after fine-tuning the batch-norm.

equalizes the effect of the training pre-processing that tends to zoom on the objects. However, increasing $K_{\text{test}}^{\text{image}}$ with $K_{\text{test}}$ fixed means looking at a smaller part of the object. This is not ideal: the object to identify is often well framed by the photographer, so the crop may show only a detail of the object or miss it altogether. Hence, in addition to increasing $K_{\text{test}}^{\text{image}}$, we *also* increase the crop size $K_{\text{test}}$ to keep the ratio $K_{\text{test}}^{\text{image}}/K_{\text{test}}$ constant. However, this means that $K_{\text{test}} > K_{\text{train}}$, which skews the activation statistics (section 3.2). The next section shows how to compensate for this skew.

### 4.2  Adjusting the statistics before spatial pooling

At this point, we have selected the "correct" test resolution for the crop but we have skewed activation statistics. Hereafter we explore two approaches to compensate for this skew.

**Parametric adaptation.** We fit the output of the average pooling layer (section 3.2) with a parametric Fréchet distribution at the original $K_{\text{train}}$ and final $K_{\text{test}}$ resolutions. Then, we define an equalization mapping from the new distribution back to the old one via a scalar transformation, and apply it as an activation function after the pooling layer (see Appendix A). This compensation provides a measurable but limited improvement on accuracy, probably because the model is too simple and does not differentiate the distributions of different components going through the pooling operator.

**Adaptation via fine-tuning.** Increasing the crop resolution at test time is effectively a domain shift. A simple way to compensate for this shift is to fine-tune the model. In our case, we fine-tune on the same training set, after switching from $K_{\text{train}}$ to $K_{\text{test}}$. We restrict the fine-tuning to the very last layers of the network. We observed in the distribution analysis that the sparsity should be adapted. This requires at least to include the batch normalization that precedes the global pooling into the fine-tuning. In this way the batch statistics are adapted to the increased resolution. We also use the test-time augmentation scheme during fine-tuning to avoid incurring further domain shifts. Figure 4 shows the pooling operator's activation statistics before and after fine-tuning. After fine-tuning the activation statistics closely resemble the train-time statistics. This hints that adaptation is successful. Yet, as discussed above, this does not imply an improvement in accuracy.

## 5  Experiments

**Benchmark data.** We experiment on the ImageNet-2012 benchmark [29], reporting validation performance as top-1 accuracy. It has been argued that this measure is sensitive to errors in the ImageNet labels [31]. However, the top-5 metrics, which is more robust, tends to saturate with modern architectures, while the top-1 accuracy is more sensitive to improvements in the model.

To assess the significance of our results, we compute the standard deviation of the top-1 accuracy: we classify the validation images, split the set into 10 folds and measure the accuracy on 9 of them, leaving one out in turn. The standard deviation of accuracy over these folds is $\sim 0.03\%$ for all settings. Therefore, we report 1 significant digit in the accuracy percentages.

We also report results for other datasets involving transfer learning in section 5.3 when presenting transfer learning results.

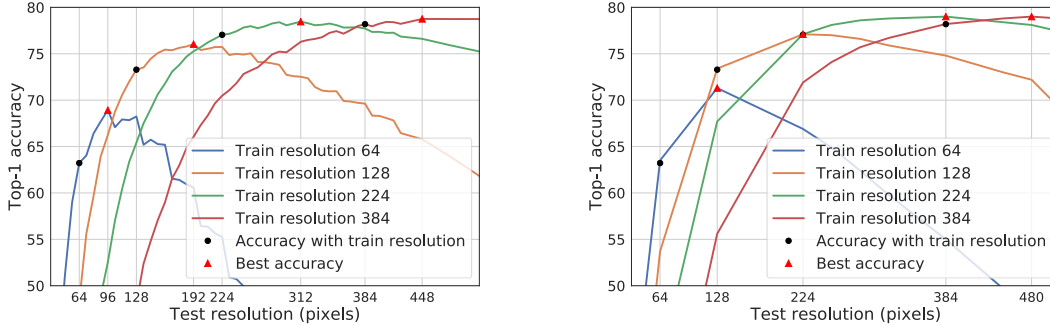

Figure 5: Top-1 accuracy of the ResNet-50 according to the test time resolution. Left: without adaptation, right: after resolution adaptation. The numerical results are reported in Appendix C. A comparison of results without random resized crop is reported in Appendix D.

**Architectures.** We use standard state-of-the-art neural network architectures with no modifications, We consider in particular ResNet-50 [11]. For larger experiments, we use PNASNet-5-Large [21], learned using *neural architecture search* as a succession of interconnected cells. It is accurate (82.9% Top-1) with relatively few parameters (86.1 M). We use also ResNeXt-101 32x48d [22], pre-trained in weakly-supervised fashion on 940 million public images with 1.5k hashtags matching with 1000 ImageNet1k synsets. It is accurate (85.4% Top-1) with lot of parameters (829 M).

**Training protocol.** We train ResNet-50 with SGD with a learning rate of $0.1 \times B/256$, where $B$ is the batch size, as in [13]. The learning rate is divided by 10 every 30 epochs. With a Repeated Augmentation of 3, an epoch processes $5005 \times 512/B$ batches, or $\sim$90% of the training images, see [5]. In the initial training, we use $B = 512$, 120 epochs and the default PyTorch data augmentation: horizontal flip, random resized crop (as in section 3) and color jittering. To finetune, the initial learning rate is $0.008$ same decay, $B = 512$, 60 epochs. For ResNeXt-101 32x48d we use the pretrained version from the PyTorch hub repository [2]. We also use a ten times smaller learning rate and a batch size two times smaller. For PNASNet-5-Large we use the pretrained version from Cadene's GitHub repository [1]. The difference with the ResNet-50 fine-tuning is that we modify the last three cells, in one epoch and with a learning rate of 0.0008. We ran our training experiments on machines with 8 Tesla V100 GPUs and 80 CPU cores.

**Fine-tuning data-augmentation.** We experimented three data-augmentation for fine-tuning: The first one (test DA) is resizing the image and then take the center crop, The second one (test DA2) is resizing the image, random horizontal shift of the center crop, horizontal flip and color jittering. The last one (train DA) is the train-time data-augmentation as described in the previous paragraph.

The test DA data-augmentation described in this paragraph is the simplest. We use it for a more direct comparison with the literature for all the results reported with ResNet-50 and PNASNet-5-Large, except in Table 2 where we report results with test DA2, which provides a slightly better performance. For ResNeXt-101 32x48d all reported results are obtained with test DA2.

Section C provides a comparison of the performance of these train-time data augmentation.

**The baseline** experiment is to increase the resolution without adaptation. Repeated augmentations already improve the default PyTorch ResNet-50 from 76.2% top-1 accuracy to 77.0%. Figure 5(left) shows that increasing the resolution at test time increases the accuracy of all our networks. E.g., the accuracy of a ResNet-50 trained at resolution 224 increases from 77.0 to 78.4 top-1 accuracy, an improvement of 1.4 percentage points. This concurs with prior findings in the literature [12].

## 5.1 Results

**Improvement of our approach on a ResNet-50.** Figure 5(right) shows the results obtained after fine-tuning the last batch norm in addition to the classifier. With fine-tuning we get the best results (79%) with the classic ResNet-50 trained at $K_{\text{train}} = 224$. Compared to when there is no fine-tuning, the $K_{\text{test}}$ at which the maximal accuracy is obtained increases from $K_{\text{test}} = 288$ to 384. If instead

Table 1: Application to larger networks: Resulting top-1 accuracy.

| Model | Train resolution | Fine-tuning | | | Test resolution | | | | | |
|---|---|---|---|---|---|---|---|---|---|---|
| | | Class. | BN | 3 cells | 331 | 384 | 395 | 416 | 448 | 480 |
| PNASNet-5-Large | 331 | – | – | – | 82.7 | 83.0 | **83.2** | 83.0 | 83.0 | 82.8 |
| PNASNet-5-Large | 331 | ✓ | ✓ | – | 82.7 | 83.4 | 83.5 | 83.4 | **83.5** | 83.4 |
| PNASNet-5-Large | 331 | ✓ | ✓ | ✓ | 82.7 | 83.3 | 83.4 | 83.5 | 83.6 | **83.7** |

| | | Class. | BN | 3 conv. | 224 | | 288 | | 320 | |
|---|---|---|---|---|---|---|---|---|---|---|
| ResNeXt-101 32x48d | 224 | ✓ | ✓ | – | 85.4 | | 86.1 | | **86.4** | |

Table 2: State of the art on ImageNet with various architectures (single Crop evaluation).

| Models | Extra Training Data | Train | Test | # Parameters | Top-1 (%) | Top-5 (%) |
|---|---|---|---|---|---|---|
| ResNet-50 Pytorch | – | 224 | 224 | 25.6M | 76.1 | 92.9 |
| ResNet-50 mix up [40] | – | 224 | 224 | 25.6M | 77.7 | 94.4 |
| ResNet-50 CutMix [39] | – | 224 | 224 | 25.6M | 78.4 | 94.1 |
| ResNet-50-D [13] | – | 224 | 224 | 25.6M | 79.3 | 94.6 |
| MultiGrain R50-AA-500 [5] | – | 224 | 500 | 25.6M | 79.4 | 94.8 |
| ResNet-50 Billion-scale [38] | ✓ | 224 | 224 | 25.6M | 81.2 | 96.0 |
| Our ResNet-50 | – | 224 | 384 | 25.6M | 79.1 | 94.6 |
| Our ResNet-50 CutMix | – | 224 | 320 | 25.6M | 79.8 | 94.9 |
| Our ResNet-50 Billion-scale@160 | ✓ | 160 | 224 | 25.6M | 81.9 | 96.1 |
| Our ResNet-50 Billion-scale@224 | ✓ | 224 | 320 | 25.6M | 82.5 | 96.6 |
| PNASNet-5 (N = 4, F = 216) [21] | – | 331 | 331 | 86.1M | 82.9 | 96.2 |
| MultiGrain PNASNet @ 500px [5] | | 331 | 500 | 86.1M | 83.6 | 96.7 |
| AmoebaNet-B (6,512) [14] | – | 480 | 480 | 577M | 84.3 | 97.0 |
| EfficientNet-B7 [34] | – | 600 | 600 | 66M | 84.4 | 97.1 |
| Our PNASNet-5 | – | 331 | 480 | 86.1M | 83.7 | 96.8 |
| ResNeXt-101 32x8d [22] | ✓ | 224 | 224 | 88M | 82.2 | 96.4 |
| ResNeXt-101 32x16d [22] | ✓ | 224 | 224 | 193M | 84.2 | 97.2 |
| ResNeXt-101 32x32d [22] | ✓ | 224 | 224 | 466M | 85.1 | 97.5 |
| ResNeXt-101 32x48d [22] | ✓ | 224 | 224 | 829M | 85.4 | 97.6 |
| Our ResNeXt-101 32x48d | ✓ | 224 | 320 | 829M | **86.4** | **98.0** |

we reduce the training resolution, $K_{train} = 128$ and testing at $K_{train} = 224$ yields 77.1% accuracy, which is above the baseline trained at full test resolution without fine-tuning.

**Application to larger networks.** The same adaptation method can be applied to any convolutional network. In Table 1 we report the result on the PNASNet-5-Large and the IG-940M-1.5k ResNeXt-101 32x48d [22]. For the PNASNet-5-Large, we found it beneficial to fine-tune more than just the batch-normalization and the classifier. Therefore, we also experiment with fine-tuning the three last cells. By increasing the resolution to $K_{test} = 480$, the accuracy increases by 1 percentage point. By combining this with an ensemble of 10 crops at test time, we obtain **83.9%** accuracy. With the ResNeXt-101 32x48d increasing the resolution to $K_{test} = 320$, the accuracy increases by 1.0 percentage point. We thus reached **86.4%** top-1 accuracy.

**Speed-accuracy trade-off.** We consider the trade-off between training time and accuracy (normalized as if it was run on 1 GPU). The full table with timings are in supplementary Section C. In the initial training stage, the forward pass is 3 to 6 times faster than the backward pass. However, during fine-tuning the ratio is inverted because the backward pass is applied only to the last layers.

In the low-resolution training regime ($K_{train} = 128$), the additional fine-tuning required by our method increases the training time from 111.8 h to 124.1 h (+11%). This is to obtain an accuracy of 77.1%, which outperforms the network trained at the native resolution of 224 in 133.9 h. We produce a fine-tuned network with $K_{test} = 384$ that obtains a higher accuracy than the network trained natively at that resolution, and the training is $2.3\times$ faster: 151.5 h instead of 348.5 h.

Table 3: Transfer learning task with our method and comparison with the state of the art. We only compare ImageNet-based transfer learning results with a single center crop for the evaluation (if available, otherwise we report the best published result) without any change in architecture compared to the one used on ImageNet. We report the Top-1 accuracy (%).

| Dataset | Models | Baseline | +our method | State-of-the-art models | |
|---------|--------|----------|-------------|-------------------------|---|
| Stanford Cars [16] | SENet-154 | 94.0 | 94.4 | EfficientNet-B7 [34] | **94.7** |
| CUB-200-2011 [36] | SENet-154 | 88.4 | **88.7** | MPN-COV [19] | **88.7** |
| Oxford 102 Flowers [23] | InceptionResNet-V2 | 95.0 | 95.7 | EfficientNet-B7 [34] | **98.8** |
| Oxford-IIIT Pets [25] | SENet-154 | 94.6 | 94.8 | AmoebaNet-B (6,512) [14] | **95.9** |
| Birdsnap [4] | SENet-154 | 83.4 | **84.3** | EfficientNet-B7 [34] | **84.3** |

**Ablation study.** We study the contribution of the different choices to the performance, limited to $K_{\text{train}} = 128$ and $K_{\text{train}} = 224$. By simply fine-tuning the classifier (the fully connected layers of ResNet-50) with test-time augmentation, we reach 78.9% in Top-1 accuracy with the classic ResNet-50 initially trained at resolution 224. The batch-norm fine-tuning and improvement in data augmentation advances it to 79.0%. The higher the difference in resolution between training and testing, the more important is batch-norm fine-tuning to adapt to the data augmentation. The full results are in the supplementary Section C.

## 5.2 Beyond the current state of the art

Table 2 compares our results with competitive methods from the literature. Our ResNet-50 is slightly worse than ResNet50-D and MultiGrain, but these do not have exactly the same architecture. On the other hand our ResNet-50 CutMix, which has a classic ResNet-50 architecture, outperforms others ResNet-50 including the slightly modified versions. Our fine-tuned PNASNet-5 outperforms the MultiGrain version. To the best of our knowledge our ResNeXt-101 32x48d surpasses all other models available in the literature. It achieves **86.4%** Top-1 accuracy and **98.0%** Top-5 accuracy, i.e., it is the first model to exceed 86.0% in Top-1 accuracy and 98.0% in Top-5 accuracy on the ImageNet-2012 benchmark [29]. This exceeds the previous state of the art [22] by 1.0% absolute in Top-1 accuracy and 0.4% Top-5 accuracy.

## 5.3 Transfer learning tasks

We have used our method in transfer learning tasks to validate its effectiveness on other dataset than ImageNet. We evaluated it on the following datasets: Stanford Cars [16], CUB-200-2011 [36], Oxford 102 Flowers [23], Oxford-IIIT Pets [25] and Birdsnap [4]. We used our method with two types of networks for transfer learning tasks: SENet-154 [3] and InceptionResNet-V2 [33]. For all these experiments, we proceed as follows. (1) we initialize our network with the weights learned on ImageNet (using models from [1]); (2) we train it entirely for several epochs at a certain resolution; (3) we fine-tune with a higher resolution the last batch norm and the fully connected layer. Table 3 reports the baseline performance and shows that our method systematically improves the performance, leading to the new state of the art for several benchmarks. We notice that our method is most effective on datasets of high-resolution images.

## 6 Conclusion

We have studied extensively the effect of using different train and test scale augmentations on the statistics of natural images and of the network's pooling activations. We have shown that, by adjusting the crop resolution and via a simple and light-weight parameter adaptation, it is possible to increase the accuracy of standard classifiers significantly, everything being equal otherwise. We have also shown that researchers waste resources when both training and testing strong networks at resolution $224 \times 224$; We introduce a method that can "fix" these networks post-facto and thus improve their performance. An open-source implementation of our method is available at `https://github.com/facebookresearch/FixRes`.

## Footnotes

[1]At training time, the extraction and resizing of the RoC is used as an opportunity to *augment* the data by randomly altering the scale of the objects, in this manner the CNN is stimulated to be invariant to a wider range of object scales.

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
