[Supplementary Material · Fixing_Resolution Supplemental.pdf]

# Supplementary material for "Fixing the train-test resolution discrepancy"

**Hugo Touvron, Andrea Vedaldi, Matthijs Douze, Hervé Jegou**

Facebook AI Research

In this supplementary material we report details and results that did not fit in the main paper. This includes the estimation of the parametric distribution of activations in Section A, a small study on border/round-off effects of the image size for a convolutional neural net in Section B and more exhaustive result tables in Section C. Section D gives additional evaluations to analyze the choice of the cropping strategy.

## A  Fitting the activations

### A.1  Parametric Fréchet model after average-pooling

In this section we derive a parametric model that fits the distribution of activations on output of the spatial pooling layer.

The output the the last convolutional layer can be well approximated with a Gaussian distribution. Then the batch-norm centers the Gaussian and reduces its variance to unit, and the ReLU replaces the negative part with 0. Thus the ReLU outputs an equal mixture of a cropped unit Gaussian and a Dirac of value 0.

The average pooling sums $n = 2 \times 2$ to $n = 14 \times 14$ of those distributions together. Assuming independence of the inputs, it can be seen as a sum of $n'$ cropped Gaussians, where $n'$ follows a discrete binomial distribution. Unfortunately, this composition of distributions is not tractable in close form.

Instead, we observed experimentally that the output distribution is close to an extreme value distribution. This is due to the fact that only the positive part of the Gaussians contributes to the output values. In an extreme value distribution that is the sum of several (arbitrary independent) distributions, the same happens: only the highest parts of those distributions contribute.

Thus, we model the statistics of activations as a Fréchet (a.k.a. inverse Weibull) distribution. This is a 2-parameter distribution whose CDF has the form:

$$P(x, \mu, \sigma) = e^{-\left(1 + \frac{\xi}{\sigma}(x-\mu)\right)^{-1/\xi}}$$

With $\xi$ a positive constant, $\mu \in \mathbb{R}, \sigma \in \mathbb{R}_+^*$. We observed that the parameter $\xi$ can be kept constant at $0.3$ to fit the distributions.

Figure 6 shows how the Fréchet model fits the empirical CDF of the distribution. The parameters were estimated using least-squares minimization, excluding the zeros, that can be considered outliers. The fit is so exact that the difference between the curves is barely visible.

To correct the discrepancy in distributions at training and test times, we compute the parameters $\mu_{\text{ref}}, \sigma_{\text{ref}}$ of the distribution observed on training images time for $K_{\text{test}} = K_{\text{train}}$. Then we increase $K_{\text{test}}$ to the target resolution and measure the parameters $\mu_0, \sigma_0$ again. Thus, the transformation is just an affine scaling, still ignoring zeros.

When running the transformed neural net on the Imagenet evaluation, we obtain accuracies:

Table 4: Matching distribution before the last Relu application to ResNet-50: Resulting top-1 accuracy % on ImageNet validation set

| Model | Train | Adapted | Fine-tuning | | Test resolution | | | | |
|---|---|---|---|---|---|---|---|---|---|
| used | resolution | Distribution | Classifier | Batch-norm | 64 | 224 | 288 | 352 | 384 |
| ResNet-50 | 224 | _ | _ | _ | 29.4 | 77.0 | 78.4 | 78.1 | 77.7 |
| ResNet-50 | 224 | ✓ | _ | _ | 29.8 | 77.0 | 77.7 | 77.3 | 76.8 |
| ResNet-50 | 224 | _ | ✓ | _ | 40.6 | 77.1 | 78.6 | 78.9 | 78.9 |
| ResNet-50 | 224 | _ | ✓ | ✓ | 41.7 | 77.1 | 78.5 | 78.9 | 79.0 |
| ResNet-50 | 224 | ✓ | ✓ | _ | 41.8 | 77.1 | 78.5 | 78.8 | 78.9 |

| $K_{\text{test}}^{\text{image}}$ | 64 | 128 | 224 | 256 | 288 | 448 |
|---|---|---|---|---|---|---|
| accuracy | 29.4 | 65.4 | 77 | 78 | 78.4 | 76.5 |

Hence, the accuracy does not improve with respect to the baseline. This can be explained by several factors: the scalar distribution model, however good it fits to the observations, is insufficient to account for the individual distributions of the activation values; just fitting the distribution may not be enough to account for the changes in behavior of the convolutional trunk.

### A.2 Gaussian model before the last ReLU activation

Following the same idea as what we did previously we looked at the distribution of activations by channel before the last ReLU according to the resolution.

We have seen that the distributions are different from one resolution to another. With higher resolutions, the mean tends to be closer to 0 and the variance tends to become smaller. By transforming the distributions before the ReLU, it is also possible to affect the sparsity of values after spatial-pooling, which was not possible with the previous analysis based on Fréchet's law. We aim at matching the distribution before the last ReLU with the distribution of training data at lower resolution. We compare the effect of this transformation before/after fine tuning with the learnt batch-norm approach. The results are summarized in Table 4.

We can see that adapting the resolution by changing the distributions is effective especially in the case of small resolutions. Nevertheless, the adaptation obtained by fine-tuning the the batch norm improves performs better in general.

## B    Border and round-off effects

Due to the complex discrete nature of convolutional layers, the accuracy is not a monotonous function of the input resolution. There is a strong dependency on the kernel sizes and strides used in the first convolutional layers. Some resolutions will not match with these parameters so we will have a part of the images margin that will not be taken into account by the convolutional layers.

In Figure 7, we show the variation in accuracy when the resolution of the crop is increased by steps of 1 pixel. Of course, it is possible to do padding but it will never be equivalent to having a resolution image adapted to the kernel and stride size.

Figure 6: Fitting of the CDF of activations with a Fréchet distribution.

Figure 7: Evolution of the top-1 accuracy of the ResNet-50 trained with resolution 224 according to the testing resolution (no finetuning). This can be considered a zoom of figure 5 with 1-pixel increments.

| test \ train | 64 | 128 | 160 | 224 | 384 | | test \ train | 64 | 128 | 224 | 384 |
|---|---|---|---|---|---|---|---|---|---|---|---|
| 64 | 63.2 | 48.3 | 40.1 | 29.4 | 12.6 | | 64 | 63.5 | 53.7 | 41.7 | 27.5 |
| 128 | **68.2** | 73.3 | 71.2 | 65.4 | 48.0 | | 128 | **71.3** | 73.4 | 67.7 | 55.7 |
| 224 | 55.3 | **75.7** | **77.3** | 77.0 | 70.5 | | 224 | 66.9 | **77.1** | 77.1 | 71.9 |
| 288 | 42.4 | 73.8 | 76.6 | **78.4** | 75.2 | | 288 | 62.4 | 76.6 | 78.6 | 75.7 |
| 384 | 23.8 | 69.6 | 73.8 | 77.7 | 78.2 | | 384 | 55.0 | 74.8 | **79.0** | 78.2 |
| 448 | 13.0 | 65.8 | 71.5 | 76.6 | **78.8** | | 448 | 49.7 | 73.0 | 78.4 | 78.8 |
| 480 | 9.7 | 63.9 | 70.2 | 75.9 | 78.7 | | 480 | 46.6 | 72.2 | 78.1 | **79.0** |

Table 5: Top-1 validation accuracy for different combinations of training and testing resolution. Left: with the standard training procedure, (no finetuning, no adaptation of the ResNet-50). Right: with our data-driven adaptation strategy and test-time augmentations.

Although the global trend is increasing, there is a lot of jitter that comes from those border effects. There is a large drop just after resolution 256. We observe the drops at each multiple of 32, they correspond to a changes in the top-level activation map's resolution. Therefore we decided to use only sizes that are multiples of 32 in the experiments.

## C  Additional result tables on Imagenet

Due to the lack of space, we report only the most important results in the main paper. In this section, we report the full result tables for several experiments.

Table 5 report the numerical results corresponding to Figure 5 in the main text. Table 6 reports the full ablation study results (see Section 5.1). Table 7 reports the runtime measurements that Section 5.1 refers to. Table 8 reports a comparaison between test DA and test DA2 that Section 5 refers to.

## D  Impact of Random Resized Crop

In this section we measure the impact of the RandomResizedCrop illustrated in the section 5. To do this we did the same experiment as in section 5 but we replaced the RandomResizedCrop with a Resize followed by a random crop with a fixed size. The figure 8 and table 9 shows our results. We can see that the effect observed in the section 5 is mainly due to the Random Resized Crop as we suggested with our analysis of the section 3.

| Train resolution | Fine-tuning | | | Test resolution (top-1 accuracy) | | | | | |
|---|---|---|---|---|---|---|---|---|---|
| | Classifier | Batch-norm | Data aug. | 64 | 128 | 224 | 288 | 384 | 448 |
| 128 | – | – | n/a | 48.3 | 73.3 | **75.7** | 73.8 | 69.6 | 65.8 |
| | ✓ | – | train DA | 52.8 | 73.3 | **77.1** | 76.3 | 73.2 | 71.7 |
| | ✓ | – | test DA | 53.3 | 73.4 | **77.1** | 76.4 | 74.4 | 72.3 |
| | ✓ | ✓ | train DA | 53.0 | 73.3 | **77.1** | 76.5 | 74.4 | 71.9 |
| | ✓ | ✓ | test DA | 53.7 | 73.4 | **77.1** | 76.6 | 74.8 | 73.0 |
| 224 | – | – | n/a | 29.4 | 65.4 | 77.0 | **78.4** | 77.7 | 76.6 |
| | ✓ | – | train DA | 39.9 | 67.5 | 77.0 | 78.6 | **78.9** | 78.0 |
| | ✓ | – | test DA | 40.6 | 67.3 | 77.1 | 78.6 | **78.9** | 77.9 |
| | ✓ | ✓ | train DA | 40.4 | 67.5 | 77.0 | 78.6 | **78.9** | 78.0 |
| | ✓ | ✓ | test DA | 41.7 | 67.7 | 77.1 | 78.6 | **79.0** | 78.4 |

Table 6: Ablation study: Accuracy when enabling or disabling some components of the training method. Train DA: training-time data augmentation during fine-tuning, test DA: test-time one.

| Resolution | | Train time per batch (ms) | | Resolution fine-tuning (ms) | | Performance | |
|---|---|---|---|---|---|---|---|
| train | test | backward | forward | backward | forward | Total time (h) | accuracy |
| 128 | 128 | 29.0 ±4.0 | 12.8 ±2.8 | – | – | 111.8 | 73.3 |
| 160 | 160 | 30.2 ±3.2 | 14.5 ±3.4 | – | – | 119.7 | 75.1 |
| 224 | 224 | 35.0 ±2.0 | 15.2 ±3.2 | – | – | 133.9 | 77.0 |
| 384 | 384 | 112.4 ±6.2 | 18.2 ±3.9 | – | – | 348.5 | 78.2 |
| 160 | 224 | 30.2 ±3.2 | 14.5 ±3.4 | – | – | 119.7 | 77.3 |
| 224 | 288 | 35.0 ±2.0 | 15.2 ±3.2 | – | – | 133.9 | 78.4 |
| 128 | 224 | 29.0 ±4.0 | 12.8 ±2.8 | 4.4 ±0.9 | 14.4 ±2.5 | 124.1 | 77.1 |
| 160 | 224 | 30.2 ±3.2 | 14.5 ±3.4 | 4.4 ±0.9 | 14.4 ±2.5 | 131.9 | 77.6 |
| 224 | 384 | 35.0 ±2.0 | 15.2 ±3.2 | 8.2 ±1.3 | 18.0 ±2.7 | 151.5 | 79.0 |

Table 7: Execution time for the training. Training and fine-tuning times are reported for a batch of size 32 for training and 64 for fine-tuning, on one GPU. Fine-tuning uses less memory than training therefore we can use larger batch size. The total time is the total time spent on both, with 120 epochs for training and 60 epochs of fine-tuning on ImageNet. Our approach corresponds to fine-tuning of the batch-norm and the classification layer.

| Models | Train | Test | Top-1 test DA (%) | Top-1 test DA2 (%) |
|---|---|---|---|---|
| ResNext-101 32x48d | 224 | 288 | 86.0 | 86.1 |
| ResNext-101 32x48d | 224 | 320 | 86.3 | 86.4 |
| ResNet-50 | 224 | 320 | 79.0 | 79.1 |
| ResNet-50 CutMix | 224 | 384 | 79.7 | 79.8 |

Table 8: Comparisons of performance between data-augmentation test DA and test DA2 in the case of fine-tuning batch-norm and classifier.

| test \ train | 64 | 128 | 224 | 384 |
|---|---|---|---|---|
| 64 | 60.0 | 48.7 | 28.1 | 11.5 |
| 96 | **61.6** | 65.0 | 50.9 | 29.8 |
| 128 | 54.2 | 70.8 | 63.5 | 46.0 |
| 160 | 42.4 | **72.4** | 69.7 | 57.0 |
| 224 | 21.7 | 69.8 | 74.6 | 68.8 |
| 256 | 15.3 | 66.4 | **75.2** | 72.1 |
| 384 | 4.3 | 44.8 | 71.7 | 76.7 |
| 440 | 2.3 | 33.6 | 67.1 | **77.0** |

Table 9: Top-1 validation accuracy for different combinations of training and testing resolution. ResNet-50 train with resize and random crop with a fixed size instead of random resized crop.

Figure 8: Top-1 accuracy of the ResNet-50 according to the test time resolution. ResNet-50 train with resize and random crop with a fixed size instead of random resized crop.