[Reviews · NeurIPS 2019]

Reviewer 1



The technical innovation of the paper is quite limited, with the exploration of an observation about the resolution discrepancy between training and testing images when performing standard data augmentation. The paper is technically sound, but the contribution is also technically very simple. The paper is clear, but it can be improved. For instance, Figures 1 and 2 are not being referred in the text. On line 42, it is not clear what the paper means by "better results". On line 74, the paper could have explained p-pooling. The results presented by the paper do not seem quite significant because just one dataset (ImageNet-2012) and two models (ResNet and PNASNet) are used. In addition, are the results shown significant? I just read authors' rebuttal for this paper and checked other reviewers' comments. Re: my review, the authors addressed them partially -- they showed results with more models, but they were vague about the significance of the results (I was expecting more formal stats significance results here). Therefore, I'm keeping my current score.

Reviewer 2



Clarity: The paper is clearly written and easy to follow. Significance: The results in the paper are significant for the practitioners and existing deployments as they shed light on the train-test resolution discrepancy and suggest method to improve test performance for existing trained models. Novelty: The analysis in this paper is novel (though improved performance on higher resolution images has been observed earlier). Questions: While the focus is on fixing discrepancy after the model has been initially trained, why not just fix the training such that there is no discrepancy, as opposed to changing the size for test and finetuning? Line 110-111 derives f = sqrt(HW), which does not seem to be right since k doesn’t include the sensor size. Typically, sqrt(hw) = 2f tan(theta_fov/2) where h, w are sensor size and not resolution, which is independent of the size. Please clarify or state what additional assumptions are made. --- After reading the author rebuttal and other reviews, I would maintain my previous rating of 8 ( a very good submission). Follow provides the key reasons 1) I like the fact that authors study and analyze an observed phenomenon that has been mentioned in passing in earlier works but never been looked at in detail. I have wondered about this myself and feel that the insights provided by this work are beneficial. 2) Authors provide a method based on their analysis that leads to test time improvements on pre-trained models. That itself is a major contribution and validation of their analysis.

Reviewer 3



Overall I think the idea is quite interesting, but because it is an empirical result, it should be validated better. Resolution difference during training and testing (mainly due to data augmentation) has long been known to the community, but few have been done to handle it. The proposed fine-tuning, albeit simple, works quite well. However, since this is an empirical submission on an empirical topic, I tend to look for more points that are "useful". For example, what is a good ratio between the data augmentation hyperparameter(s) and the train/test size ratio? In other words, a general rule of thumb for practioners in image recognition on the ImageNet dataset. And, are the observations generalizable? If I transfer the ImageNet model to a specific task (e.g., CUB), will the findings in this paper be useful? If the answer is yes, how can it be useful? How about other domains where scale difference is not caused by data augmentation? e.g., object detection? ----- I raised my recommended score a bit after reading the response. The author response answered most of my questions (except the detection one), and the answer to R1 concerning state-of-the-art (86.4%) on ImageNet is interesting.

[Author Response · NeurIPS 2019]

# Author response for "Fixing the train-test resolution discrepancy"

We thank the reviewers for their constructive feedback on the paper. We will take into account their comments on presentation and typos. Here we answer their main questions and comments.

**R1: The results presented by the paper do not seem quite significant because just one dataset (ImageNet-2012) and two models (ResNet and PNASNet) are used. In addition, are the results shown significant?** We will add experiments on additional models and datasets.

In particular, we have evaluated our approach for transfer learning for low-resource and/or fine-grained classification. Following common practice, (1) we initialize the model with weights obtained on ImageNet and (2) fine-tune it on the new dataset, which provides the baseline performance. Then (3) we use our method, i.e. we fine-tune the last batch norm and the fully connected layer with a higher resolution. Table 1 shows that for all transfer-learning tasks our method obtains a gain in accuracy of +0.4% to +1.3% absolute. All those results are obtained on competitive benchmarks where tens to hundreds of researchers have evaluated their methods, gains of around 1% accuracy are therefore significant.

Finally, we applied our method to a very large ResNeXt-101 32x48d from [Mahajan et al. ECCV'18], available online. We improved their top-1 Imagenet accuracy from 85.4% to 86.4%, which is the new state of the art on ImageNet.

**R2: why not just fix the training such that there is no discrepancy, as opposed to changing the size for test and finetuning?** We have tried other approaches to fix the discrepancy, which are not discussed for space reasons. The approach we presented consistently achieved the best performance while being simple and reducing the training time.

Regarding R2's proposal, since the discrepancy is mainly due to the random resized crop data augmentation, we replaced it with a fixed size random crop and resize to $K_\text{train}$=224 pixel. When testing at $K_\text{test}$ =224 we obtain 74.6% accuracy. The best test resolution is $K_\text{test}$ =256: 75.2% accuracy, so there is an improvement at a slightly higher resolution. However, the result is significantly below the baseline accuracy obtained with random resize crop and without any resolution change (76.2%). This shows that it is important to keep a distinct data augmentation between train and test.

**R2: Line 110-111 derives f = sqrt(HW), which does not seem to be right since k doesn't include the sensor size.** As often done for compactness (see [Hartley & Zisserman] eq. 5.9 or 6.9 depending on the edition), we expressed the focal length in unit of pixels; namely, the camera projection equation is $x = W_\text{pixels}^\text{sensor}/W_\text{mm}^\text{sensor} f_\text{mm} X/Z$ which simplifies to $x = fX/Z$ by setting $f = f_\text{pixels} = W_\text{pixels}^\text{sensor}/W_\text{mm}^\text{sensor} f_\text{mm}$. We will clarify this in the final version.

**R3: what is a good ratio between the data augmentation hyperparameter(s) and the train/test size ratio? In other words, a general rule of thumb for practioners in image recognition on the ImageNet dataset.** Thank you, we will add a "best practices" paragraph. It is hard to define a single ratio that works for all resolutions because the difference in statistics between the activations in the neural network does not vary linearly, eg. because of the padding of convolutional layers. An important effect to consider is the interaction between the input resolution and the size of the feature map before spatial pooling. There are key resolutions (largest size for a given feature map size) that provides local performance minima. For example, for a resnet-50 this occurs each time the resolution is a multiple of 32. Hence, a practical approach is to try out a few resolution steps above the training resolution, ie. $\{256, 288, .., 384\}$. This approach applies both with and without fine-tuning, but fine-tuning improves the effect.

**R3: And, are the observations generalizable? If I transfer the ImageNet model to a specific task (e.g., CUB), will the findings in this paper be useful? If the answer is yes, how can it be useful?** Yes, please see our answer to R1.

| Dataset | Model | Baseline | Ours | State-Of-The-Art Models | | |
|---|---|---|---|---|---|---|
| iNaturalist 2017 | SENet-154 | 74.1 | **75.4** | IncResNet-V2-SE | [Horn *et al.* ArXiv'17] | 67.3 |
| Stanford Cars | SENet-154 | 94.0 | 94.4 | EfficientNet-B7 | [Tan & Le, ArXiv'19] | **94.7** |
| CUB-200-2011 | SENet-154 | 88.4 | **88.7** | MPN-COV | [Peihua Li *et al.* ArXiv'19] | 88.7 |
| Oxford 102 Flowers | IncResNet-V2 | 95.0 | 95.7 | EfficientNet-B7 | [Tan & Le, ArXiv'19] | **98.8** |
| Oxford-IIIT Pets | SENet-154 | 94.6 | 94.8 | AmoebaNet-B (6,512) | [Yanping Huang *et al.* ArXiv'18] | **95.9** |
| NABirds | SENet-154 | 88.3 | **89.2** | PC-DenseNet-161 | [Dubey *et al.* ArXiv'17] | 82.8 |
| Birdsnap | SENet-154 | 83.4 | **84.3** | EfficientNet-B7 | [Tan & Le, ArXiv'19] | **84.3** |

Table 1: Transfer learning tasks. Evaluation is top-1, single crop and without changing the base Imagenet architecture.

[Meta-Review · NeurIPS 2019]

I think the paper addresses an interesting problem, albeit limited in scope to computer vision. I am sure practitioners in that field will appreciate the paper's findings. Two of the reviewers were positive, and reaffirmed their position during the post-rebuttal discussion, while R1 remained concerned, in particular regarding lack of rigorous statistical analysis of the results. The other reviewers did not consider that issue a deal-breaker, and I agree and recommend to accept.